# Functional and Dysfunctional Neuroplasticity in Learning to Cope with Stress

**DOI:** 10.3390/brainsci10020127

**Published:** 2020-02-24

**Authors:** Simona Cabib, Paolo Campus, David Conversi, Cristina Orsini, Stefano Puglisi-Allegra

**Affiliations:** 1Department of Psychology, University of Rome ‘La Sapienza’, 00185 Rome, Italy; david.conversi@uniroma1.it (D.C.); cristina.orsini@uniroma1.it (C.O.); stefano.puglisi-allegra@uniroma1.it (S.P.-A.); 2Department of Experimental Neurosciences, IRCCS Fondazione Santa Lucia, 00143 Rome, Italy; 3Department of Psychiatry and Molecular and Behavioral Neuroscience Institute, University of Michigan, Ann Arbor, MI 48109, USA; pcampus@umich.edu; 4IRCCS Neuromed, via Atinense 18, 86077 Pozzilli (IS), Italy

**Keywords:** active avoidance, epigenetics, extinction learning, genetic models, glucocorticoid, hippocampus, learned helplessness, post-traumatic stress disorder

## Abstract

In this brief review, we present evidence of the primary role of learning-associated plasticity in the development of either adaptive or maladaptive coping strategies. Successful interactions with novel stressors foster plasticity within the neural circuits supporting acquisition, consolidation, retrieval, and extinction of instrumental learning leading to development of a rich repertoire of flexible and context-specific adaptive coping responses, whereas prolonged or repeated exposure to inescapable/uncontrollable stressors fosters dysfunctional plasticity within the learning circuits leading to perseverant and inflexible maladaptive coping strategies. Finally, the results collected using an animal model of genotype-specific coping styles indicate the engagement of different molecular networks and the opposite direction of stress effects (reduced vs. enhanced gene expression) in stressed animals, as well as different behavioral alterations, in line with differences in the symptoms profile associated with post-traumatic stress disorder.

## 1. Introduction

Converging evidence points to the involvement of learning-related dysfunctional neuroplasticity in psychopathology [1,2,3,4,5]. Moreover, preclinical and clinical findings converge in supporting development of the therapeutic interventions capable of influencing learning-related neuroplasticity [6,7,8,9,10]. Maladaptive learning through stress experiences is the candidate mediator of this involvement. Indeed, stress is a recognized non-genetic cause of behavioral disturbances; it is known to influence learning, and behavioral adaptation to stress requires learning. Learning through stress is required for functional adaptation to unexpected changes of the physical, social, and physiological environment, because it allows developing and stabilizing effective coping strategies. On the other hand, gene–experience interactions can influence learning processes, leading to the development of maladaptive coping. In the present review, we consider neuroplasticity supporting either adaptive or maladaptive stress coping.

The view on stress as a dynamic interaction between potentially dangerous or adverse experiences and the behavioral, cognitive, and physiological responses that these experiences elicit from organisms allows us to discriminate between adaptive and maladaptive coping. When organisms appraise a novel experience as potentially dangerous/adverse, they express species-typical defensive responses such as fight, flight, and hide (or freeze), as well as previously acquired strategies to avoid, remove, or control the situation. If the individual defensive repertoire proves ineffective, organisms appraise the experience as stressful, i.e., demanding beyond their actual means [3,11,12,13,14,15]. Coping develops from action-oriented and intrapsychic efforts to manage stressors and terminate or moderate stress responses, thus preventing allostatic load [16]. Indeed, appraisal of an experience as stressful is associated with very high levels of emotional arousal, and with a stereotypic pattern of physiological responses, such as release of the corticotropin-releasing hormone, adrenocorticotropic hormone, and corticosterone/cortisol. Although these responses support the organism’s ability to sustain stress experiences, they are physiologically expensive and dangerous in the long run [16]. For instance, high blood pressure can trigger myocardial infarction and accelerate atherosclerosis, whereas a moderately elevated serum cortisol concentration inhibits bone formation, while persistent activation of the sympathetic system and of the hypothalamic–pituitary–adrenal axis fosters weight loss, amenorrhea, and anorexia nervosa [17]. Therefore, once activated, physiological stress responses have to be terminated as soon as possible [3,15,16].

Newly developed coping strategies appraised as capable of reducing physiological and emotional arousal are learned, thus becoming part of the individual’s repertoire [3,15]. Human coping strategies are numerous, but they can be grouped into two broad categories: those targeting the stressor (problem-focused) and those targeting the emotional arousal that sustains stress responses (emotion-focused) [11,12]. Exposure to novel stressors also fosters two classes of coping strategies in laboratory animals depending on specific characteristics of the stressors. Exposure to stressors that can be removed, avoided, escaped, or just controlled promotes development and stabilization of specific defensive responses (freezing, escape, avoidance, and species-typical social displays), whereas stressors that prove insensitive to these responses promote helplessness, i.e., no attempts to deal with the source of stress. These two strategies are known as active and passive coping, respectively, and are strongly reminiscent of human problem-focused and emotion-focused coping [18,19,20,21].

## 2. Adaptive Coping and Functional Neuroplasticity

Both active and passive coping strategies are adaptive, because learning rules control their stabilization as long-term memories so that they can be easily retrieved when needed, and rapidly dismissed when ineffective. Thus, active coping strategies are acquired as goal-directed instrumental responses that undergo extinction whenever the goal (control/escape) is not available, whereas passive coping is acquired through extinction learning that is strongly context-specific and less persistent in time than instrumental learning.

Active avoidance is the prototypic model of active coping, and although the role of instrumental learning in active avoidance paradigms has been a matter of debate, recent evidence supports the view that conditioned avoidance responses depend on both Pavlovian and instrumental conditioning [22,23]. In line with this view, neuroplasticity markers have been detected within the brain circuit in different phases of active avoidance learning the corticostriatal circuit involved in goal-directed instrumental learning [24,25,26,27].

Active avoidance training increases expression of c-Fos in the shell substructure of the nucleus accumbens (NAcSh) [28], expression of the early growth response protein 1 (Egr1) in the orbitofrontal and dorsal anterior cingulate cortex (CG1), and in the hippocampal CA1 region [29]. It also fosters expression of the activity-regulated cytoskeleton-associated protein (Arc/Arg3.1) in the dentate gyrus (DG) of the hippocampus [29]. Retrieval of the acquired avoidance enhances expression of Egr1 in the DG and increases c-Fos expression in neurons of the prefrontal cortex (pFC) projecting toward the basolateral amygdala (BLA) and the ventral striatum (VS) [28,29]. Extinction of active avoidance increases c-Fos expression in neurons of the pFC projecting toward the BLA and the VS, as well as brain-derived neurotrophic factor (BDNF) expression in the ventral hippocampus (vHPC) [28,30]. Finally, rats showing the highest density of Arc/Arg3.1-expressing neurons in the CG1 also show the highest number of avoidances and the fastest escape latencies supporting the primary role of cortical plasticity in the acquisition of active avoidance [29]; either silencing neural activity in the NAcSh or disconnecting the basal amygdala (BA) and the NAcSh disrupts acquired avoidance indicating the need for functional connectivity between the BA and the NAcSh for the retrieval of the active avoidance memory [31], and reducing BDNF production in vHPC neurons impairs the recall of avoidance extinction indicating a role of vHPC plasticity in the long-term memory of extinction [30].

The prototypic model of passive coping is the immobility determined using the forced swimming test (FST), also known as the Porsolt test after the researcher who developed it [32]. During the first FST experiment (10 to 15 min), animals vigorously swim around and struggle to climb the container’s walls. As the water tank for the FST is small, does not have a safe platform, and is inescapable, these responses decrease overtime, whereas episodes of immobility (only small movements required to keep the head above the water) increase in frequency and duration. When re-exposed to the FST, rats and mice immediately express high levels of immobility; indeed, the response is consolidated as a long-term memory [33,34,35]. Immobility was used as a measure of depressive-like behavior; nonetheless, there is now wide consensus on the view that this behavioral response is an adaptive passive coping strategy in a stressful situation that cannot be avoided nor escaped [3,36,37,38,39,40]. Indeed, immobility prevents useless and risky loss of energy making it available if a change in the situation requires it.

As described above, the coping response to the FST is dynamic, and different brain circuits are involved not only in acquisition and consolidation of passive coping, but also in sustaining initial attempts to active coping and in shifting to passive coping. A pathway connecting the prelimbic region (PL) of the pFC with the bed nucleus of the stria terminalis restrains passive coping responses in novel stressful situations [41], and specific input from the medial part of pFC (mpFC) to the dorsal raphe nucleus (DRN) is selectively inhibited when rats shift from active behavior to immobility in the FST [42]. Moreover, the manipulations that prevent stress-induced serotonin (5-HT) release in the mpFC also prevent development of immobility through disruption of the serial pFC-BLA connectivity [36,43], and the manipulations that prevent stress-induced dopamine (DA) release in the mpFC selectively prevent reduction of DA availability in the nucleus accumbens (NAc), as well as development of helplessness [44,45,46].

The neuroplasticity required to acquire immobility, to consolidate it as a long-term memory, and to recall it on subsequent exposure to the stressor is mediated by stress hormones through activation of mineralocorticoid receptors (MRs) and glucocorticoid receptors (GRs) in different brain regions [38]. Data on neuroplasticity associated with different phases of learning in the FST are rare because of the diffusion of the single FST protocol. Nonetheless, compelling evidence of the specificity of learning-relevant neuroplasticity comes from the findings that previous regular voluntary exercise increases the immobility expressed on the test day without affecting the behavioral response to the first FST experience. Regular voluntary exercise also enhances FST-induced histone H3 phospho-acetylation and c-Fos expression in the granule neurons of the DG [47]—an effect mediated by activation of NMDA receptors and of the extracellular signal-regulated kinase (ERK) 1⁄2 ⁄ mitogen- and stress-activated kinase (MSK) 1⁄2 pathway in mature granule neurons [48], possibly by activation of local GRs [49].

Both consolidation and retrieval of the acquired passive coping strategy require circulating corticosterone. Indeed, adrenalectomy before the FST prevents expression of immobility checked for 24 h later, and this deficit is eliminated by the corticosterone administered both immediately after training and before the testing [33]. Moreover, temporary inactivation of either the dorsal hippocampus or of the infralimbic (IL) component of the mpFC immediately after the first FST experience reduces retrieval of immobility 24 h later indicating the involvement of these brain areas in the consolidation of the acquired passive coping [34,50]. An active IL component is selectively required for the extinction of the Pavlovian associative and of goal-directed learning, including active avoidance [28,51]. Thus, the role played by the IL component in the stabilization of passive coping as a long-term memory supports a homology between extinction learning and acquisition of a passive coping strategy [34,50] in agreement with previous findings [52].

## 3. Maladaptive Coping

Overgeneralization, inflexibility, and perseveration resulting from experience or from the interaction between genotype and experience characterize maladaptive coping.

The classic model of experience-dependent maladaptive coping is learned helplessness. Learned helplessness is a generalization of acquired passive coping to novel potentially escapable stressful situations, and is fostered by prolonged experiences with inescapable/uncontrollable stressors. The model was developed by Martin E.P. Seligman and Steven F. Maier in the early 1960s (for a recent review, see [53]) for dogs and humans. Later on, Maier perfected a rat protocol known as the “triadic” protocol, because it involves triplets of animals: a “shocked” rat that receives trains of shock at random intervals and can temporarily interrupt shock delivery by an operant response—the escapable stress (ES) condition; a “yoked” rat that is connected in parallel with the shocked one, thus receiving the same amount of shock for the same time, but being unable to exert control on shock delivery—the inescapable stress (IS) condition; and a rat exposed to the same apparatus without receiving shock—the “control” condition. IS rats, but not ES ones, were unable to learn an escape response in a context radically different from that used for the triadic training, and this deficit lasted as long as three days [20] indicating generalization of helplessness to novel stressful experiences. In more recent experiments, it was demonstrated that repeated defeat fosters IS-like behavioral and neural effects [54], whereas IS fosters social avoidance [55], extending the concept of learned helplessness beyond the escape/avoidance deficit.

Using the triadic protocol in rats, Maier and his colleagues were able to identify the DRN as the main node of a brain circuit activated by the IS and inhibited by the ES. Most interestingly, they demonstrated inhibition of the DRN 5-HT neurons by the ES mediated by excitatory inputs from the mpFC to intrinsic GABAergic neurons that leads to a rapid reduction of 5-HT release induced by shock experience [53]. On the other hand, the learned helplessness syndrome is due to sensitization of the DRN 5-HT release dependent on desensitization of local 5-HT1A inhibitory auto-receptors [53]. It should be pointed out that although both the maladaptive coping and the dysfunctional neuroplasticity fostered by the IS are temporary, they disrupt the individual’s ability to deal with novel avoidable/controllable stressors activating pathogenic sequelae.

An individual bias towards the expression of passive or active coping responses in the presence of the same novel stressor has been observed in murine genetic models. Thus, the rats selectively bred for rapid acquisition of the active avoidance behavior (Roman High Avoidance, RHA) show persistent active coping in the FST, particularly in the recall test, in comparison with the rats that are slow active avoidance learners (Roman Low Avoidance, RLA) [56,57,58,59]. Moreover, the mice genetically selected for short attack latency (SAL) are characterized by prolonged active coping in the FST and high levels of active avoidance, whereas the long attack latency (LAL) mice show rapid expression of passive coping in the FST and low avoidance learning [13,60,61,62]. Finally, mice of the genetically unrelated C57BL/6 (B6) and DBA/2 (D2) inbred strain show high and low levels of immobility, respectively, in the FST [46,63], and the D2 mice outperform the B6 mice in the protocols involving active avoidance or escape learning [64,65,66]. These findings are very relevant, because they support genetic determinants of coping styles beyond laboratories and species.

Although many neurobiological phenotypes were investigated in each genetic model, research on the B6 vs. D2 model was the most focused on the neurobiology of stress coping [3]. Mice of the B6 strain show a rapid decrease of the NAc DA during the first FST experience in comparison with the D2 mice. This strain difference depends on a larger FST-induced DA release in the mpFC of the B6 mice; thus, reduction of the mpFC DA response induced by different means resulted in a slower decrease of the DA in the NAc, as well as in a reduction of immobility in the B6 mice [63]. Moreover, when exposed to a novel inescapable/uncontrollable stress, the B6 mice displayed higher 5-HT outflow in the mpFC and higher GABA outflow in the BA than the D2 mice [36,67]. In addition, in the baseline conditions, the B6 mice are characterized by a higher number of 5-HT1A receptors in the pFC and a lower number of GABAB receptors in the BA than the D2 mice [67,68]. In light of the previously discussed mediators of brain responses to escapable/controllable and inescapable/uncontrollable stressors, these strain-specific neurobiological phenotypes are coherent with a bias in the passive vs. active coping by the B6 and the D2 mice, respectively.

## 4. Stress-Induced Dysfunctional Neuroplasticity and Psychopathology

It is worth pointing out that a bias toward the use of a specific coping strategy is not a risk marker for maladaptive coping. Thus, individual coping styles have been reported and studied in humans, and they are considered psychological constructs that share features with constitutional variables, and are associated with variability in the frontocortical components of the anterior salience network (a large-scale brain network that is primarily composed of the anterior insula and the dorsal anterior cingulate cortex) [69]. However, only the B6 mice develop learned helplessness following the IS [55,70], as well as persistent helplessness following chronic stress [71]. Because the inability to deal with adverse experiences by means of active (or problem-focused) coping strategies is considered a marker of a psychopathology [72,73,74,75], these findings seem to indicate that the B6 mice represent a model of genetic liability to stress. In line with this view, in the B6 mice, but not in mice of the D2 strain, a previous experience of an inescapable shock fosters extinction-resistant conditioned freezing [55]—the classic model of post-traumatic stress disorder (PTSD) in rodents [76,77].

However, perseverant active avoidance is also a recognized symptom of PTSD, and an active avoidance insensitive to contingency degradation characterizes obsessive–compulsive disorder (OCD) patients [26,78,79]. Moreover, behavioral perseveration despite adverse consequences is the main marker of pathological compulsivity [80], and chronically stressed D2 mice relapse into active coping when tested for helplessness retention in the FST [50], and strain-specifically develop activity-based anorexia (ABA), running on wheels instead of feeding to the point of dying of starvation if the protocol is not interrupted [81,82]. Finally, drug addiction involves compulsivity [80] and chronically or repeatedly stressed D2 mice strain-specifically develop behavioral sensitization to psychostimulants [83,84], a phenotype typically fostered by prolonged exposure to addictive drugs and supported by addiction-related aberrant neuroplasticity [85,86]. PTSD is characterized by comorbidity with different psychiatric conditions including drug abuse, behavioral addictions, anxiety disorders, major depression, and OCD [87,88,89,90,91], and liability to a specific comorbidity can be genotype-dependent [77]. Thus, the reviewed results suggest that stress-induced dysfunctional neuroplasticity within brain circuits supporting individual coping styles are responsible for specific PTSD-associated disturbances.

The use of mice from well-known inbred strains allows exploiting a wealth of literature on the biological phenotypes involved in the strain-specific maladaptive coping that characterizes the B6 and D2 mice. Strain differences for the acute corticosterone response were observed, depending on the stressor. Thus, a faster peak increase was observed in the B6 mice following exploration of a novel environment [92]; the opposite was observed following immobilization within a rotating restraining apparatus, or training for step-through inhibitory avoidance [93,94]. Moreover, no strain difference was observed following 60 min of restraint [95]; a greater response by the D2 mice following 15 min of restraint [96], and a greater response by the B6 mice following 10 min of the FST (first experience) [97]. On the other hand, repeated restraint increased FST-induced plasma corticosterone level in the mice of both strains [97], whereas chronic mild stress (CMS) did not influence the corticosterone response to an acute stress challenge in either strain [96]. It should be pointed out, however, that the D2 mice are characterized by higher availability of both GRs and MRs in the hippocampus [93,94], and that the same increase in the circulating corticosterone has different effects on neuroplasticity in the mice of the B6 and D2 strains [93,98,99].

Two studies have focused on genome-wide gene expression analysis in the mice of the two strains following repeated restraint or CMS stress protocols and an acute challenge with a different stressor. Although one study was performed on males and the other one on females using different stress paradigms, they both found striking strain differences [96,97]. Mozhui et al. [97] found highly divergent gene expression changes in the mpFC, the amygdala, and the hippocampus between male B6 and D2 mice, with very few genes showing alterations in both strains in any region (2–10 depending on the region). Thus, the authors concluded that the mice of the two strains differ in terms of the gene networks engaged by stress rather than in terms of the degree of activation of the common molecular “stress network”. Terenina et al. [96] reported that nearly all of the common transcripts detected in the hippocampus of the stressed female mice were increased in the B6 mice and reduced in the D2 mice. Some of these genes were enriched for synaptic localization leading the authors to conclude that they act in a genotype-dependent manner to modulate hippocampal neuronal excitability in response to stress.

In line with these strain-specific effects of stress on gene expression, temporary (12 days) food restriction (FR) reduces the c-Fos expression promoted by the first FST experience in the dorsolateral striatum (DLS), the NAcSh and the IL of male D2 mice, whereas it increases the c-Fos expression in the amygdala, the hippocampal CA1 region, and the IL of male B6 mice [37,50]. Consolidation of FST-induced helplessness requires a functioning hippocampus in mice of the B6 strain and a functioning DLS in the D2 mice, but is dependent on a functioning IL in both strains [34,50]. Finally, FR-exposed mice of the B6 strain show long-lasting helplessness [71], whereas FR D2 mice show relapse into active coping already 24 h after the first exposure to the FST [50,100]. Together, these findings support the view that FR strengthens memory consolidation of a passive coping strategy in the B6 mice, whereas it disrupts it in the D2 mice by altering brain plasticity induced by learning through stress in a genotype-specific way. It should be pointed out that FR reduces the number of DA receptors of the D2 type in the DLS of the D2 mice, a phenotype typically observed in addicted people, and a pharmacological blockade of D2 receptors replicates effects of restricted feeding in free-feeding mice of this inbred strain [100]. The latter finding offers strong support to the homology between the stress-induced and the addiction-associated dysfunctional neuroplasticity. In this regard, it is worth pointing out that enhanced corticosterone is required for cocaine-induced behavioral sensitization and neuroplasticity only in mice of the D2 strain [93,98,99].

Finally, most PTSD individuals are hypervigilant to avoid exposure to threatening stimuli, whereas patients with the less common dissociative subtype of PTSD often exhibit passive defensive responses [101,102]. Converging evidence from human and animal studies supports the view that activation of the dorsolateral periaqueductal gray (PAGdl) is associated with expression of active coping, whereas activation of the ventrolateral PAG (PAGvl) has been associated with passive coping [101,102,103,104], and recent findings indicate that PTSD patients of the dissociative type exhibit greater functional connectivity of the PAGvl [102]. In the course of an exploratory investigation, we looked for brain areas where c-Fos expression correlated with that of the PAGdl and/or the PAGvl in FR mice of the B6 and D2 strains. Results are presented in Figure 1. The most striking strain difference was absence of significant correlations with the PAGdl c-Fos expression in mice of the B6 strain. In FR D2 mice, instead, correlated c-Fos expression traced a circuit involving both components of the periaqueductal gray (PAG). These findings are coherent with the perseverant passive coping expressed by FR mice of the B6 strain, support the hypothesis that different PAG-centered circuits mediate development of behavioral disturbances following traumatic experiences, and further stress the value of rodent genetic models of coping styles for the understanding of pathogenic outcomes of learning through stress.

## 5. Conclusions

In this short review, we summarized evidence indicating the primary role of dysfunctional neuroplasticity fostered by learning through stress in the development of psychiatric disturbances. Learning through stress is needed for development of a rich individual repertoire of adaptive coping responses and neuroplasticity associated with instrumental goal-directed learning and extinction learning supports acquisition and consolidation of active and passive coping, respectively. Adaptive coping is flexible and context-specific, as are the two types of learning processes involved. However, prolonged or repeated experiences with ISs can alter the learning processes leading to generalized and inflexible coping responses. Moreover, genetically determined coping styles have been identified in animal models that rely on different brain circuits to support learning in stressful situations. Finally, the study of dysfunctional neuroplasticity fostered by pathogenic stressors in these models can reveal molecular determinants of specific pathological outcomes.

## Figures and Tables

**Figure 1 brainsci-10-00127-f001:**
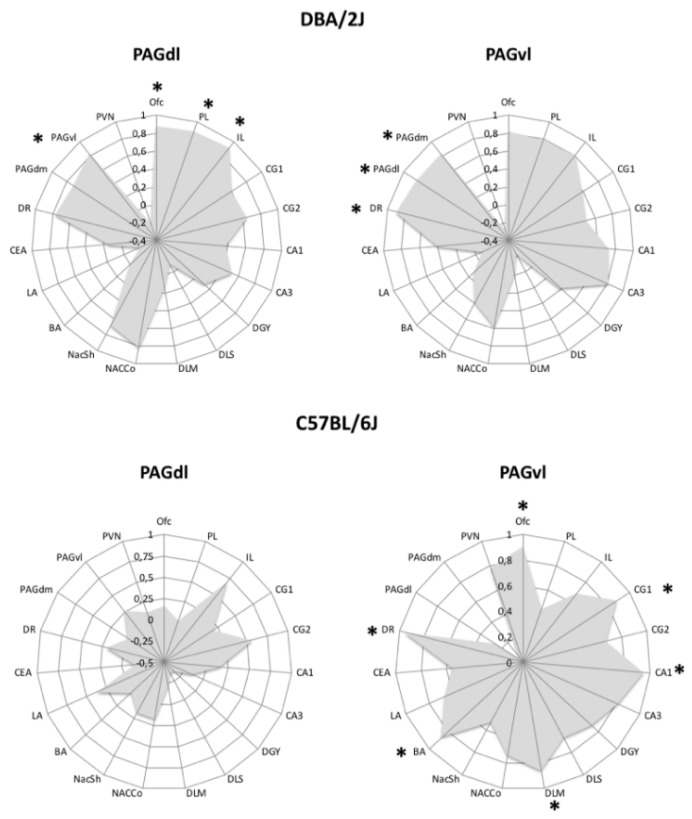
Twelve days of restricted feeding are associated with different c-Fos positive cell number correlation profiles of the PAGdl and the PAGvl in mice of the D2 (top) and B6 (bottom) strains. Radar graphs depict Pearson’s *r* values. Regions with c-Fos expression significantly correlated with expression in the target PAG subdivision are indicated by an asterisk (* = *p* < 0.05). Abbreviations: Ofc = orbitofrontal cortex; PL = prelimbic vortex; IL = infralimbic cortex; CG1 = dorsal anterior cingulate cortex; CG2 = ventral anterior cingulate cortex; CA1, CA3, DGY = hippocampus; DLS = dorsolateral striatum; DMS = dorsomedial striatum; NAcCo = nucleus accumbens core; NAcSh = nucleus accumbens shell; BA = basal Amygdala; LA = lateral amygdala; CEA = central amygdala; DR = dorsal raphe; PAGdl = dorsolateral periaqueductal gray; PAGdm = dorsomedial PAG; PAGvl = ventrolateral PAG; PVN = paraventricular nucleus of hypothalamus.

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
