# Peer review of "Functional and Dysfunctional Neuroplasticity in Learning to Cope with Stress"

_brainsci, 2020, doi:10.3390/brainsci10020127_

Round 1

Reviewer 1 Report

The authors addressed the highly relevant topic of

The authors addressed the highly relevant topic of learning-related plasticity of adaptive or maladaptive coping strategies with emphasis on the role of stress in dysfunctional plasticity. Active and passive strategies are emphasized in animals models. My suggestions are as follows:

I suggest to include a short section dealing with the aspects of the article in relation to humans from a translational perspective. Please comment about studies in humans targeting learning related plasticity and maladaptive learning specially with neuromodulation approaches:

Pedroarena-Leal, N.; Heidemeyer, L.; Trenado, C.; Ruge, D. Human Depotentiation following Induction of Spike Timing Dependent Plasticity. Biomedicines 2018, 6, 71.

Hays, S. A., Rennaker, R. L., & Kilgard, M. P. (2013). Targeting plasticity with vagus nerve stimulation to treat neurological disease. Progress in brain research, 207, 275–299. doi:10.1016/B978-0-444-63327-9.00010-2

Active and passive strategies are emphasized in animals models. My suggestions are as follows:

I suggest to include a short section dealing with the aspects of the article in relation to humans from a translational perspective. Please comment about studies in humans targeting learning related plasticity and maladaptive learning specially with neuromodulation approaches:

Pedroarena-Leal, N.; Heidemeyer, L.; Trenado, C.; Ruge, D. Human Depotentiation following Induction of Spike Timing Dependent Plasticity. Biomedicines 2018, 6, 71.

Hays, S. A., Rennaker, R. L., & Kilgard, M. P. (2013). Targeting plasticity with vagus nerve stimulation to treat neurological disease. Progress in brain research, 207, 275–299. doi:10.1016/B978-0-444-63327-9.00010-2

Author Response

We thank the reviewer fo the helpful suggestions that are part of the revised version of the MS (p. 1, lines 25-28)

Reviewer 2 Report

In this brief review, the authors delivered an exquisite and well-written manuscript in which they discussed evidence concerning functional and dysfunctional learning-based neuroplasticity that occurs during stress coping strategies. Most ideas were presented in clear and easy-to-digest ways, and these ideas are highly relevant and accordant within the field. My overall enthusiasm for this manuscript is high. Yet, the authors need to address some concerns.

Concerns:

- For lines #45-46, “Although these responses support the organism’s ability to sustain stress experiences, they are physiologically expensive and dangerous in the long run…”, please explain, expand or provide example(s) for why these responses are “dangerous in the long run”.

- In lines #58-60 (end of introduction), this sentence seems “out of the blue”, random and very unanticipated, and does not flow well with the rest of the paragraph and introduction. I suggest deletion of this sentence.

- In lines #68-73, the authors argue that: (a) “…although the role of instrumental learning in active avoidance paradigms has been a matter of debate, findings from recent studies… offer strong support to this conclusion”, and (b) “Indeed, active avoidance involves prefrontal cortex (pFC), amygdala, nucleus accumbens (NAc), and hippocampus; in other words, the cortico-striatal circuit involved in goal-directed instrumental learning.” The fact that active avoidance and instrumental learning involve similar brain regions and circuits does not necessarily mean that instrumental learning plays a role in active avoidance. In fact, these brain regions are involved in a plethora of functions, including fear learning, extinction, reward learning, habitual behavior, anxiety, drug addiction, observational learning, etc, etc. Yet, this does not mean interactions among all these functions. Can the authors paraphrase these sentences, or discuss more compelling evidence?

- In line #191, define “anterior salience network”.

- In line #218, not sure what is the meaning of “restraint +rotation”.

- In line #224, define “Bmax” for a broader audience to understand.

- In Figure 1 legend, define/explain “absolute value of r”. What values were used to calculate r? As it is, it is not clear what is being correlated.

Typos:

- In line #30, revise grammar; missing a “to” (“physiological environment because it allows “to” develop and stabilize…”).

- In line #214, revise grammar; perhaps missing a “to” (“The use of mice from well-known inbred strains allows “to” exploit a wealth of literature…).

Author Response

We thank you for the careful reading and helpful suggestions that we have included in the new version of the MS as it follows.

1) we added discussion, explanations, as  well as references where required:

p.2: lines 50-54; 75-79; 

2) we removed the last sentence of the introduction

3) we corrected useless and awkward scientific jargon

p.4: lines 23-25; p.5: lines 253-254; 260;

4) corrected typos

best regards  
